# Difference in Levels of Vitamin D between Indoor and Outdoor Athletes: A Systematic Review and Meta-Analysis

**DOI:** 10.3390/ijms24087584

**Published:** 2023-04-20

**Authors:** Maria Bârsan, Vlad-Florin Chelaru, Armand-Gabriel Râjnoveanu, Ștefan Lucian Popa, Andreea-Iulia Socaciu, Andrei-Vlad Bădulescu

**Affiliations:** 1Department of Occupational Medicine, ‘Iuliu Hațieganu’ University of Medicine and Pharmacy Cluj-Napoca, 400347 Cluj-Napoca, Romania; maria.opritoiu@umfcluj.ro (M.B.);; 2Faculty of Medicine, ‘Iuliu Hațieganu’ University of Medicine and Pharmacy Cluj-Napoca, 400347 Cluj-Napoca, Romania; badulescu.andreivlad@elearn.umfcluj.ro; 32nd Medical Department, ‘Iuliu Hațieganu’ University of Medicine and Pharmacy Cluj-Napoca, 400347 Cluj-Napoca, Romania

**Keywords:** vitamin D, indoor activity, outdoor activity, athletes

## Abstract

Vitamin D, its importance in different processes taking place in the human body, the effects of abnormal levels of this hormone, either too low or too high, and the need for supplementation have been extensively researched thus far. Variances in exposure to sunlight can cause vitamin D levels to fluctuate. Indoor activity can be a factor for these fluctuations and can lead to a decrease in vitamin D levels. We conducted a systematic review and meta-analysis aiming to identify whether indoor compared to outdoor training has a significant influence on vitamin D levels; we also performed subgroup analyses and multivariate meta-regression. The type of training has an impact on vitamin D levels that is influenced by multiple cofounders. In a subgroup analysis not considering cofounders, the mean serum vitamin D was 3.73 ng/mL higher in outdoor athletes, a difference which barely fails to achieve significance (*p* = 0.052, a total sample size of 5150). The indoor–outdoor difference is only significant (clinically and statistically) when considering studies performed exclusively on Asian athletes (a mean difference of 9.85 ng/mL, *p* < 0.01, and a total sample size of 303). When performing the analyses within each season, no significant differences are observed between indoor and outdoor athletes. To control for multiple cofounders (the season, latitude, and Asian/Caucasian race) simultaneously, we constructed a multivariate meta-regression model, which estimated a serum vitamin D concentration lower by 4.446 ng/mL in indoor athletes. While a multivariate model suggests that outdoor training is associated with slightly higher vitamin D concentrations when controlling for the season, latitude, and Asian/Caucasian race, the type of training has a numerically and clinically small impact. This suggests that vitamin D levels and the need for supplementation should not be decided based on training type alone.

## 1. Introduction

For the human body, vitamin D is an essential fat-soluble hormone with multiple functions, the main one being the regulation of calcium and phosphate levels [1]. This and other functions can be explained by the fact that this hormone regulates the activity of target genes that control varied cellular mechanism functioning in the metabolism of xenobiotics [2], the skin [3], and the immune [4] and cardiovascular systems [5].

The synthesis of vitamin D is influenced by the amount of sunlight (ultraviolet B radiation, 290–315 nm wavelength) on exposed skin. Before it can begin its activity, it must be metabolically activated by hydroxylation, a process that takes place in the liver and subsequently in the kidneys and is modulated by multiple factors, such as 1,25(OH)2D, the phosphate and calcium levels in the blood, and the parathyroid hormone [6]. The amount of ultraviolet B radiation that a person is exposed to varies according to the zenith angle, air pollution, and altitude. Factors regarding the properties of the skin can also cause differences in the synthesis of vitamin D: the use of sunscreen (SPF 30 or higher), skin pigment, and age. A negative impact on circulating levels could be caused by indoor activity, life at high latitudes [7], and other environmental or occupational factors [8], in addition to the well-known deficit in intake, whereas hypervitaminosis D is recognized as being an effect of a too high intake of vitamin D-fortified food and prescribing or dispensing vitamin D supplements in an inappropriate matter [9].

Optimal levels, with increasing requirements due to aging, are situated at 50–100 nmol/L, with low values linked to osteomalacia in adults and rickets in children but also increased risk of death caused by cancer, cardiovascular disease [10], and other conditions [11]. Furthermore, there are also studies linking low levels of vitamin D to poor mental health [12] and even depression [13]. Current research surprisingly contradicts earlier data, suggesting that cancer incidence and mortality are not causally associated with 25-hydroxyvitamin D [14]. Even when it comes to the supplementation of vitamin D, it only seems to be advisable in order to maintain physiological levels in patients suffering from hypovitaminosis D [15], as achieving levels of vitamin D that are too high can also carry its risks [16].

Due to its many functions, especially those engaged in maintaining bone, muscle, and immune health, vitamin D plays a very important role in an athlete’s performance [17]. Current research mentions the prevalence of vitamin D insufficiency as high in athletes [18,19]. This can be of paramount importance, seeing as an excess of exercise can have a negative effect on immunity [20]. Observations such as these lead to the development of recommendations regarding the monitoring of the vitamin D level in athletes, adjusting lifestyle and diet, and even supplementation if required [21,22,23].

Considering the aforementioned importance of vitamin D for an athlete’s quality of life and performance and that its level is closely linked to sun exposure, we designed a study aimed at identifying if practicing indoor sports as opposed to outdoor sports can significantly impact vitamin D levels. Due to the fact that vitamin D levels are significantly influenced by factors other than training type, we also investigated the impact of the season, self-reported race, latitude, and gender, and whether the type of training remains an independent predictor of serum vitamin D when accounting for these variables.

## 2. Materials and Methods

We conducted a systematic review aiming to identify whether there were any differences in vitamin D levels among athletes practicing indoors or outdoors. This study was performed in accordance with Preferred Reported Items for Systematic Reviews and Meta-Analysis (PRISMA) [24,25,26]. We have registered our study with PROSPERO (CRD42022384047). The review question was “How does practicing indoor or outdoor physical activity impact vitamin D levels in athletes?” and its outcome was the mean vitamin D level in athletes practicing indoor or outdoor sports. Additionally, we performed subgroup analysis to account for the influence of self-reported race and the season during which vitamin D levels were measured. To better account for cofounders, gender, season, self-reported race, and latitude were included in multivariate meta-regression models.

As very few studies investigated the difference between vitamin D levels in indoor and outdoor athletes directly compared to the relative abundance of studies of vitamin D levels in athletes practicing a specific sport, we proceeded with a meta-analysis of means, rather than mean differences, on which we then performed subgroup analyses.

### 2.1. Article Eligibility: Inclusion and Exclusion Criteria

We included articles and dissertations indexed by the queried databases and returned by our search strategies, for which the full text was available, either in English or German or for which an English version was available. All articles had to be published between 1 January 2012 and 31 December 2022.

We considered all observational studies as eligible, including retrospective or prospective cohorts, as well as cross-sectional studies. Interventional studies were only considered eligible if they included baseline measurements before any intervention was applied.

Reasons for the exclusion of articles were:Articles of type: case reports, case series, or secondary literature (reviews, meta-analysis, book chapters);Articles for which full text was impossible to source: conference abstracts and posters and unpublished data;Retracted articles;Patients under treatment for which pre-treatment data were unavailable;Patients suffering from illness for which pre-illness data were unavailable;Studies completed in silico, in vitro, or in vivo on nonhuman subjects.

### 2.2. Search Strategy

We have searched the following electronic databases: PubMed, Embase, Scopus, ScienceDirect, ProQuest, and Lilacs, as well as Cochrane Central Register of Controlled Trials, ClinicalTrials.gov, International Clinical Trials Registry Platform (of the WHO) and EBSCO Open Dissertations and Katalog der Deutschen Nationalbibliothek. The searches were performed in January 2023. We analyzed articles published in English or German or for those with English or German versions available. Only articles published and available in the mentioned databases between 1 January 2012 and 31 December 2022 were taken into consideration.

Search strategies were established following a PICO model: population of study (athletes excluding athletes suffering from illness or under treatment for which no baseline data was available), intervention (indoor sports), comparison (outdoor sports), and outcome (vitamin D mean). The search query used for each database is available in Appendix A.

### 2.3. Data Collection

One author (AVB for Katalog der Deutschen Nationalbibliothek and EBSCO Open Dissertations) and VFC for the other databases executed the searches in the databases. Results were downloaded in available standard formats and then centralized using Clarivate EndNote, which was then used to remove duplicates. Remaining information about each article, meaning its title, authors, and web link, was introduced in a common database and then randomly assigned to two blinded authors for the screening of the title. Results were compared between the authors, and differences in opinion were solved by discussion. The remaining articles were then assigned to two authors (VFC and AVB) for abstract and full-text screening, as well as quality assessment and data extraction. Disagreements were solved by a third author (MB). All authors agreed with the final articles included in the meta-analysis.

Vitamin D concentrations that were expressed in nmol/L were converted to ng/mL in order to allow pooling.

### 2.4. Quality Assessment

Quality assessment was undertaken by two blinded authors (AVB and VFC), and disagreements were solved by discussion. The Newcastle–Ottawa scale [27] was used for cross-sectional studies and randomized clinical trials, and the NHLBI scale [28] was used for uncontrolled before-and-after studies.

In terms of the scales’ design, the NOS tools have sections dedicated to selection, comparability, and either exposure or outcome; however, the maximum score for each section differs between scales [27]. Subsequently, comparisons between the quality of studies of different types are not possible. Moreover, there is no unanimously defined threshold below which a study is considered at high risk for bias.

### 2.5. Data Analysis

Data extraction was undertaken by two blinded authors (VFC, AVB) following a standardized form and collecting the following information: mean, standard deviation, and sample size, as well as type of training (outdoor/indoor) and the following optional fields: gender, season (out of 4 possible: spring, summer, autumn, and winter), race (as a proxy for skin color, which was rarely reported consistently in the included studies), and latitude. For articles that reported subgroup-level data, we extracted data pertaining to the smallest nonoverlapping subgroups available. In order to avoid the ecological fallacy [29,30], analyses of categorical variables were not performed using percentages but only on homogeneous subsets; specifically, when studying the effect of gender, we only included samples comprising entirely of athletes of the same gender and tested them via subgroup analyses and dummy variable meta-regression, rather than using the percentage of female athletes as a predictor in meta-regressions.

All statistical analyses were performed using R (R Foundation) [31] with RStudio (Posit Software, PBC, Boston, MA, USA) [32], with the packages *meta* [33] and *metafor* [34].

We assessed heterogeneity using the I^2^ value [35], defining an I^2^ value over 75% as significant heterogeneity, and the DerSimonian-Laird τ^2^ estimate [36]. In all cases, we identified significant heterogeneity; therefore, random-effects estimates were computed. The use of random-effects methods is also supported by the existence of highly influential covariates, such as season or skin color, which differed considerably between studies. Subgroup differences were evaluated using a χ^2^ fixed-effects test proposed by Borenstein and Higgins [37] and then by meta-regression.

Meta-regressions were performed following a mixed-effects, standard-error-weighted least squares model, using restricted maximum likelihood (REML) fitting. Similar to an ordinary least squares regression model, the equation of a mixed-effects meta-regression takes on the following form:$${\hat{\theta }}_{k}=\theta +\beta x_{k}+\varepsilon_{k}+\zeta_{k}$$
where ${\hat{\theta }}_{k}$ is the estimated effect size (in our case, the mean vitamin D concentration), *θ* is an intercept term signifying the effect size for $x_{k}=0$, β is a fixed-effects coefficient interpreted as the regression slope, $\varepsilon_{k}$ is the sampling error within each individual study, and $\zeta_{k}$ is the random-effects error indicating the existence of heterogeneity between studies. R^2^ has a similar interpretation to linear regression, i.e., it can be interpreted as the fraction of variability that is explained by the predictor, while the test of moderators is an ANOVA omnibus test for the whole model and is analogous to the F-test in linear regression [38].

Small-study bias was evaluated by visual examination of funnel plots, as well as by means of Egger’s test [39], which is recommended by the Cochrane Handbook [29] for meta-analyses of at least 10 studies, which was the case in the present study, in all subgroups.

The significance threshold was 0.05 for all tests.

## 3. Results

### 3.1. Study Identification and Selection

The database search identified 709 published articles and 38 clinical trial protocols. After removing duplicates, title screening was performed, which removed articles without the study topic or not matching the inclusion criteria. A total of 100 articles remained for abstract and full-text screening, and 33 articles remained in the final meta-analysis. Figure 1 presents the PRISMA flowchart for the selection of articles.

### 3.2. Quality Assessment and Risk-of-Bias Evaluation

We identified 21 cross-sectional studies, 10 cohort studies, 1 randomized controlled trial (RCT), and 1 before-and after study without a control group (Table 1).

For cross-sectional studies, the risk of bias was generally high, with a median score of 5 (of 10) and a maximum of 7 (in only two studies), with the main issues referring to sample selection. Specifically, only four studies were awarded at least 1 point (of 3) in this section: three received the point for sample representativeness (as most included studies provided very unclear, if any, description of the sampling strategy), one article justified its sample size by power analysis, and one study reported a low non-respondent rate (of the two studies that reported response rates). One study (i.e., that of Mehran et al. [52]) received 2 points for sampling for both representativeness and a satisfactory non-response rate. The assessment of outcomes was another item with poor quality; only three studies relied on self-reported outcomes (awarded 1 point out of a maximum of 2) and two studies reporting blinded outcome assessments. Very few studies had issues regarding the choice of statistical tests, comparability, or exposure assessments.

Similar deficiencies were observed for cohort studies: 2 of 10 articles received a point for cohort representativeness; regarding outcome assessments, 2 of 12 articles used self-reported outcomes, and none used blinded assessment. The comparability of the cohorts was mostly satisfactory (eight studies controlled for at least one factor), and other items were generally without issues; overall, a median score of 6.5 of 9 was obtained.

The one identified RCT, that of Wyon et al. [71], had an unclear definition of controls, ascertainment of exposure, and non-response rate, leading to a score of 6 (of 9), respectively.

The issues encountered in the single identified before-and-after study [72] refer to selection bias (not all eligible patients were included, and the sample size was not justified) and outcome (the outcome assessment was not blinded, the subjects were evaluated for the outcome only once, and individual-level data were not reported), leading to an overall score of 7 of 12.

The complete results for all articles, including scores for each item, are presented in Appendix A.

### 3.3. Evaluation of Publication Bias

Publication bias was evaluated by means of funnel plots and funnel plot asymmetry testing (i.e., Egger’s test). While relatively symmetrical, the funnel plot for all included studies (Figure 2) shows widely scattered observations: even large studies with small standard errors tend to obtain rather different results.

In order to verify whether heterogeneity in the study conditions obscures the presence of publication bias, we recreated the funnel plot and performed Egger’s test for studies grouped by season and type of sport (indoor or outdoor sport). The results are presented in Appendix A, and the plots are provided in Appendix A. While the plots show a similar distribution of results, they remain relatively symmetrical. Subsequently, in all cases, we failed to prove the existence of significant small-study bias.

**Figure 2 ijms-24-07584-f002:**
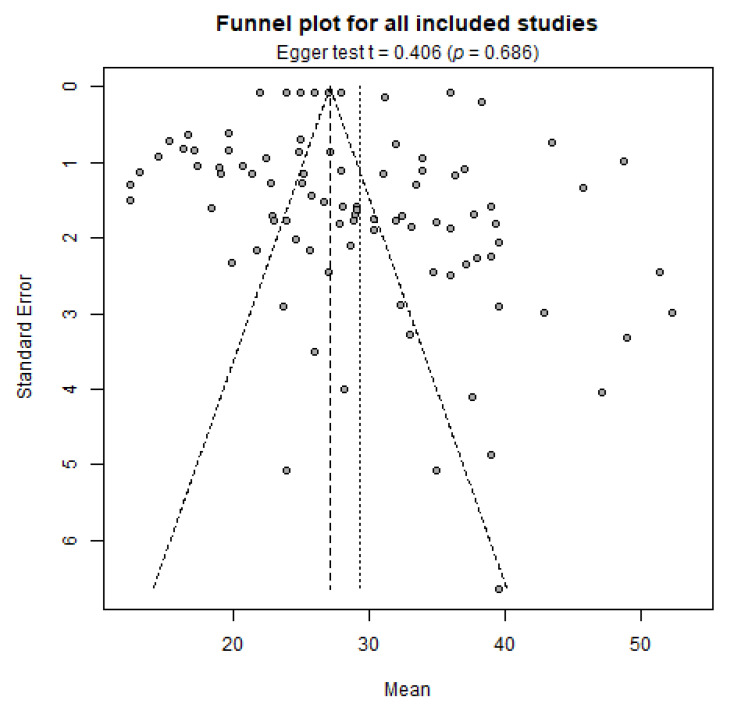
Funnel plot for all identified studies.

### 3.4. Pooling and Forest Plots

To elucidate the differential influence of indoor and outdoor training on serum vitamin D concentration, we performed random-effects pooling after defining subgroups based on the type of sport. As meta-analysis software does not allow for nested subgroups, we first subsetted for the variables which we had previously identified as possible confounders, namely season and race.

We first performed pooling on all available observations (93 samples from 33 studies, total sample size n = 5150), obtaining a pooled mean vitamin D concentration of 29.32 ng/mL (95% CI: 27.49–31.14). The values show a very high level of heterogeneity (I^2^ = 100%, τ^2^ = 76.55, *p* < 0.001).

After subsetting according to the type of training, we identified a numerically larger vitamin D concentration in the outdoor group (31.4 ng/mL, 95% CI: 28.56–34.24, 40 samples from 20 studies, sample size n_out_ = 2734) compared to the indoor group (27.67 ng/mL, 95% CI: 25.38–29.97, 53 samples from 25 studies, total sample size n_in_ = 2416), leading to a mean difference of 3.73 ng/mL in favor of the outdoor group. The fixed-effects test for the subgroup differences suggests a trend toward significance (statistic = 4.00, df = 1, *p* = 0.052). The forest plot is provided in Appendix A due to its large size (Appendix A).

Nonetheless, even after subsetting for the type of training, both the outdoor (I^2^ = 100%, τ^2^ = 81.31, *p* < 0.001) and the indoor (I^2^ = 99%, τ^2^ = 67.81, *p* < 0.001) groups suffer from very high heterogeneity.

In order to exclude the effect of cofounders and attempt to decrease the observed heterogeneity, we repeated the pooling and recreated the forest plots for the following subsets: studies performed in spring, summer, autumn, and winter, and, respectively, studies performed on African-American, Asian, and Caucasian athletes; and contrasting studies performed on indoor and outdoor athletes. The forest plots are provided in Figure 3, Figure 4, Figure 5, Figure 6, Figure 7, Figure 8 and Figure 9.

Subgroup analyses for Asian athletes (Figure 3) show a significantly higher serum vitamin D concentration in outdoor athletes (34.91 ng/mL, 95% CI 30.60–39.22, 7 samples from 3 studies, sample size n_out_ = 122) compared to indoor athletes (25.06 ng/mL, 95% CI 21.04–29.08, 8 samples from 4 studies, sample size n_in_ = 181), for a mean difference of 9.85 ng/mL (*p* < 0.01, total sample size n = 303). Seasonality was comparable in the two subgroups, with one exception: Kim et al. [48] recruited indoor athletes at unspecified time points over 4 years. The research of Huang et al. [45] was performed near the equator and is minimally affected by seasonality.

**Figure 4 ijms-24-07584-f004:**
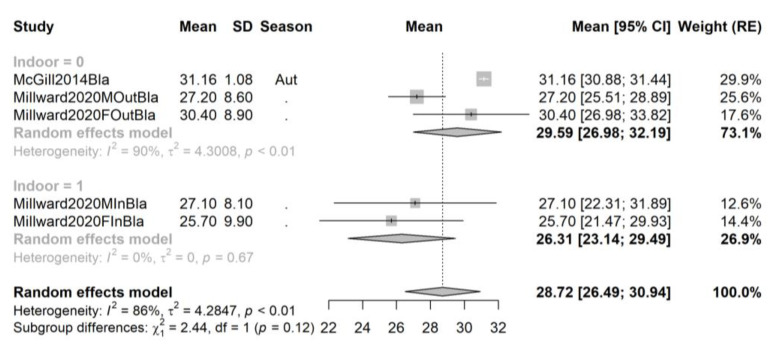
Forest plot of all studies performed on African-American athletes sorted by season. Studies [51,68].

Very few available studies (five samples from two studies) provided data for African-American athletes (Figure 4). While there is a slightly higher vitamin D concentration in the outdoor group (29.59, 95% CI 26.98–32.19, 3 samples from 2 studies, sample size n_out_ = 183) compared to the indoor group (28.72, 95% CI 26.49–30.94, 2 samples from 1 study, sample size n_in_ = 32), the two confidence intervals are wide and overlapping, and the subgroup test fails to achieve significance (*p* = 0.12) partially due to the small pooled sample size in the indoor group.

In the case of Caucasian athletes (Figure 5), while the number of studies and the pooled sample size are significantly higher (39 samples from 23 studies, total sample size n = 3530), seasonality is rather different between the indoor and outdoor group, with a relatively higher proportion of spring and winter studies in the indoor group, which may obscure the actual difference in the means associated with the type of training. Once again, there is a trend in favor of outdoor training (mean 32.38 ng/mL, 95% CI 27.19–37.58, 17 samples from 12 studies, sample size n_out_ = 1792) compared to indoor training (27.65, 95% CI 23.53–31.77, 22 samples from 14 studies, sample size n_in_ = 1738), without reaching significance (mean difference = 4.73, *p* = 0.16).

**Figure 5 ijms-24-07584-f005:**
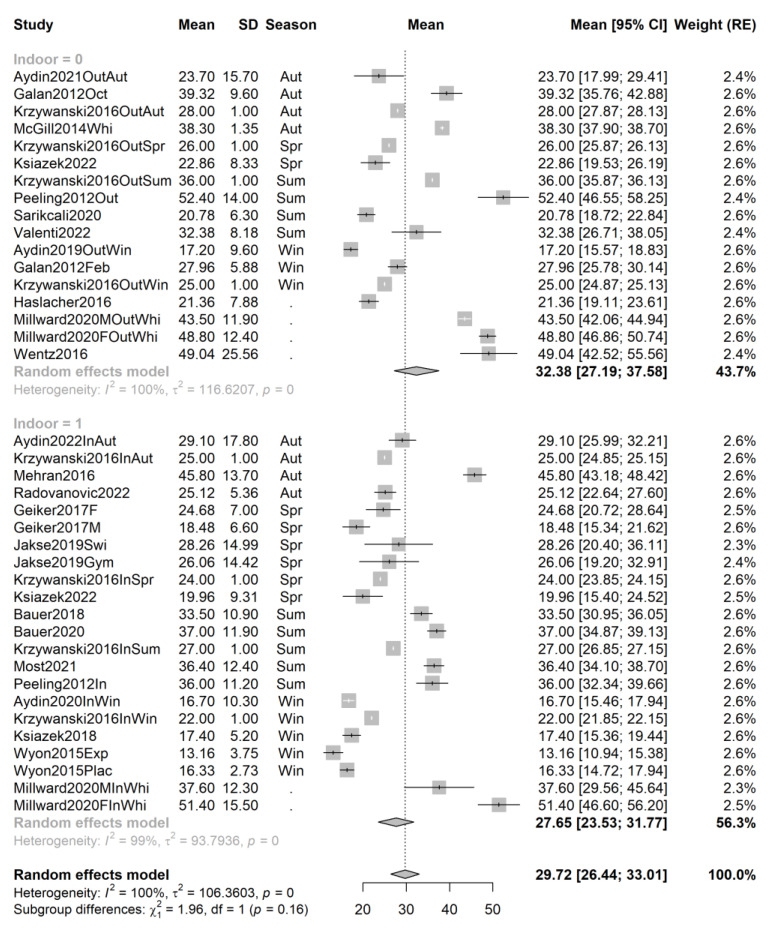
Forest plot of all studies performed on Caucasian athletes sorted by season. Studies [40,41,42,44,46,49,50,51,52,53,54,55,57,60,63,64,66,68,71,72].

Performing the analyses within groups defined by self-reported race has led to a slight reduction in heterogeneity for the studies of Asian athletes (I^2^ values of 95 and 96% for outdoor and indoor studies, respectively, compared to 100 and 99% in the overall analysis) but not for Caucasian athlete studies; heterogeneity is markedly reduced for African-American studies, but this can be attributed to the fact that very few studies remain for analysis.

**Figure 6 ijms-24-07584-f006:**
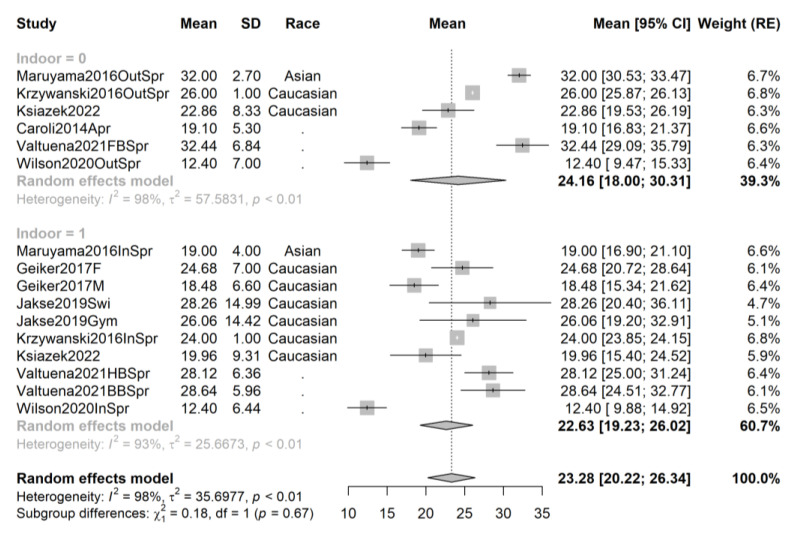
Forest plot of all studies performed in spring sorted by athletes’ race. Studies [44,46,50,61,66,67,69,70].

When analyzing studies grouped by season, we notice the lack of significant differences, even at the 0.1 threshold of significance, between indoor and outdoor athletes and an important overlap of the 95% confidence intervals. Compared to the previous analyses, the trend in favor of outdoor athletes is negligible and even reversed for studies performed in autumn.

For spring studies (16 samples from 8 studies, total sample size n = 642), there is a minimal mean difference of 0.88 ng/mL in favor of outdoor athletes (mean 24.16 ng/mL, 95% CI 18.00–30.31, 6 samples from 6 studies, sample size n_out_ = 325) compared to indoor athletes (mean 22.63, 95% CI 19.23–26.02, 10 samples from 7 studies, sample size n_in_ = 317), which is not statistically significant (*p* = 0.67).

**Figure 7 ijms-24-07584-f007:**
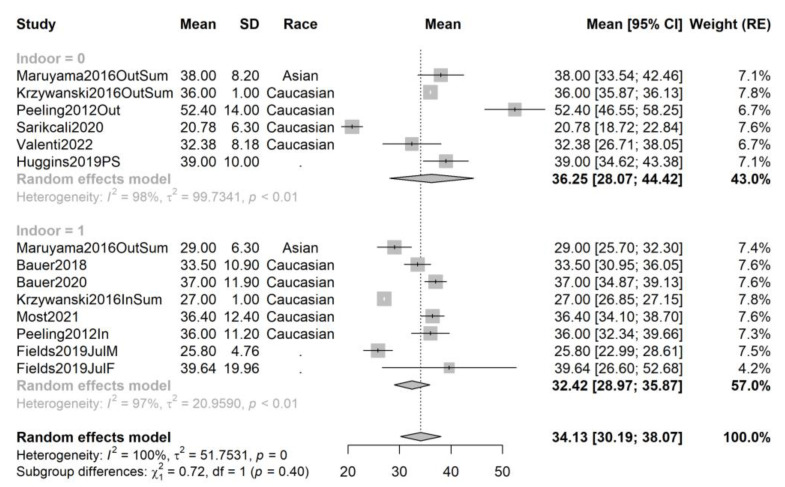
Forest plot of all studies performed in summer sorted by athletes’ race. Studies [54,57,65,66,67,72].

For summer studies (14 samples from 10 studies, total sample size n = 878), the mean serum vitamin D is higher by 3.83 ng/mL in outdoor athletes (mean 36.25, 95% CI 28.07–44.42, 6 samples from 6 studies, sample size n_out_ = 328) compared to indoor athletes (mean 32.42, 95% CI 28.97–35.87, 8 samples from 7 studies, sample size n_in_ = 550). Nonetheless, this fails to reach statistical significance (*p* = 0.40).

**Figure 8 ijms-24-07584-f008:**
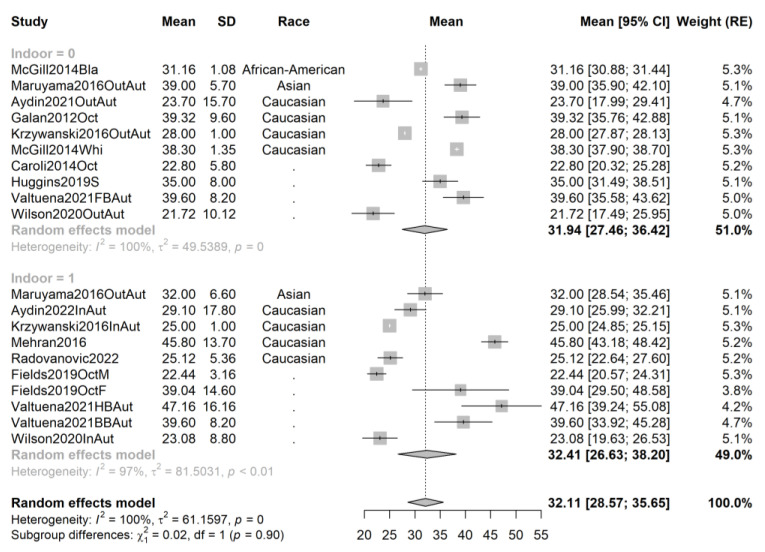
Forest plot of all studies performed in autumn sorted by athletes’ race. Studies [40,43,51,52,55,61,63,65,66,67,69,70].

Contrary to expectations, serum vitamin D is (marginally) higher (mean difference 0.44 ng/mL, *p* = 0.9) in indoor athletes in autumn studies (20 samples from 12 studies, total sample size n = 989). There were nine outdoor studies, providing 10 samples and n_out_ = 479 observations; eight indoor studies were available, from which we extracted 10 samples and n_in_ = 510 observations.

**Figure 9 ijms-24-07584-f009:**
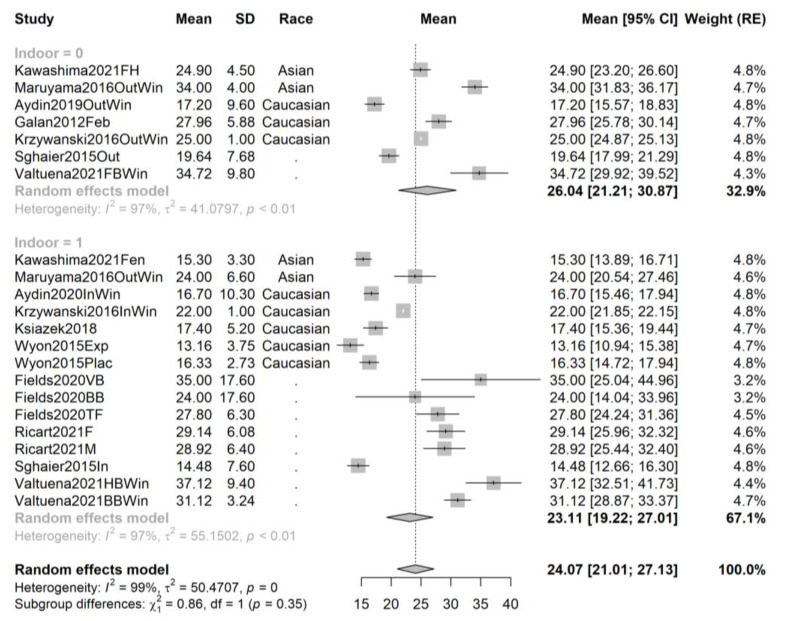
Forest plot of all studies performed in winter sorted by athletes’ race. Studies [40,47,49,56,58,62,63,66,67,70,71].

For winter studies (22 samples from 11 studies, total sample size n = 1210), the mean serum vitamin D is higher by 2.93 ng/mL in outdoor athletes (n_out_ = 529 observations from 7 samples and 7 studies) compared to indoor athletes (15 samples from 10 studies, n_in_ = 681). However, the subgroup test shows no significant differences (*p* = 0.35)

### 3.5. Meta-Regression

Taking into account the low power of subgroup tests and the inability to evaluate the effect of cofounders, we created a number of meta-regression models, both bivariate and multivariate.

Five data points were available for African-American athletes obtained as subgroups from only two studies (McGill (2014) [51] and Millward et al. (2020) [68]); subsequently, in the race-subsetted forest plots, we only pooled studies performed on Asian and Caucasian athletes.

In the bivariate models, the only significant predictors of vitamin D levels are latitude and the type of sport. However, the R^2^ values are rather small (10.52 and 3.69%, respectively), indicating that they are only responsible for a minor share of the variation in vitamin D levels.

The results of each bivariate regression are presented in Table 2 below. Due to the presence of missing data for either variable, the number of studies used to compute each regression model differs between the models. As the results of the test for moderators (see Materials and Methods) are identical to those of the *t*-test for the slope coefficient, they are not included in this table. Significant predictors are marked with an asterisk in the first column, while the intercept is significant in all models.

The effect of season was investigated by means of multivariate regression. In this case, the regression equation has the following form:$${\hat{\theta }}_{k}=\theta +\beta_{Spr}D_{Spr}+\beta_{Sum}D_{Sum}+\beta_{Win}D_{Win}+\varepsilon_{k}+\zeta_{k}$$
where $D_{Spr},D_{Sum},D_{Win}$ all take either 0 or 1; if $D_{Spr}=D_{Sum}=D_{Win}=0$, then *θ* represents the mean vitamin D level of athletes in the autumn studies.

The coefficients of spring and winter are highly significant, indicating that vitamin D levels for athletes measured in spring and winter tend to be significantly lower compared to the ones in autumn. Summer has a slight positive effect on vitamin D levels but fails to reach statistical significance. Nonetheless, the R^2^ value is rather high, indicating that 27.88% of the variance in vitamin D levels is explained by seasonality. In total, 72 studies were included, and the results are presented below in Table 3. Statistically significant results are marked with an asterisk.

Finally, a multivariate model encompassing indoor/outdoor sport, season, and latitude was created, as described in the equation below:$${\hat{\theta }}_{k}=\theta +\beta_{Indoor}D_{Indoor}+\beta_{Spr}D_{Spr}+\beta_{Sum}D_{Sum}+\beta_{Win}D_{Win}+\beta_{Latitude}\cdot Latitude+\varepsilon_{k}+\zeta_{k}$$

The results (see Table 4) indicate that the type of sport is not an independent predictor of vitamin D levels after controlling for latitude and season, while seasonality (specifically, spring and winter relative to autumn, similar to the model above) and latitude retain significance. While the model explains a satisfactory part of the variance in vitamin D levels (R^2^ = 31.06%), it is a relatively small improvement compared to the season-only model and is also rather complex, with five coefficients, apart from the intercept, applied on a relatively small number of observations (n = 66).

Considering the observations highlighted in Figure 3 and Figure 5, we contemplated a model which included race (Caucasian or Asian) as a predictor (Table 5). In this case, the type of sport becomes a significant predictor. However, the sample size is noticeably smaller (n = 40), and the risk of overfitting is considerable, with 6.67 observations per predictor other than the intercept. The R^2^ value is rather high (52.98%), but may be inflated due to the large number of predictors and, implicitly, overfitting. Moreover, the impact of seasonality becomes less obvious, with winter being the only predictor which is significant at a 0.05 level of confidence.

## 4. Discussion

Vitamin D is involved in multiple aspects of the metabolism and the workings of the human body, in the regulation of calcium levels, parathyroid hormone and calcitonin production, and bone mineral density, as well as in innate immunity [73], respiratory infections prevention [74], pregnancy and miscarriage [75,76], and thyroid dysfunctions [77,78], among others. Athletes undertaking intense physical activity are at risk of stress fractures [79], for which vitamin D supplementation can prove protective. Moreover, at least for lower limb muscle power, vitamin D supplementation can have a positive effect, especially for athletes training indoors [80]. A previous meta-analysis found that 56% of athletes had inadequate vitamin D levels, and indoor training, especially in higher latitudes, was a risk factor for vitamin D deficiency [81].

To date, our meta-analysis is the first to quantify the impact of indoor/outdoor training on serum vitamin D levels and complements the work of Farrokhyar et al. [81]. While Farrokhyar et al. investigate the prevalence of vitamin D insufficiency and deficiency in indoor and outdoor athletes, our study attempts to estimate the quantitative effect (more specifically, its magnitude) of outdoor training on vitamin D levels. Our research has revealed a complex landscape in that its significance is greatly dependent on other confounders, of which season, race, and latitude seem the most important.

When ignoring the possibility of confounding, the effect of the type of training is statistically significant, albeit small in magnitude (indoor athletes have, on average, vitamin D levels which are lower by 3.692 ng/mL in a bivariate dummy variable regression model); the amount of variance in vitamin D levels explained by indoor/outdoor training is likewise minimal (R^2^ = 3.69%).

When taking into account multiple predictors, the effects of the type of training seem to be dependent on latitude, race, and season: indoor training is associated with a vitamin D level lower by 4.446 ng/mL in a model incorporating the above-mentioned predictors. However, this observation stems from a multivariate regression model with 6 predictors (aside from the intercept) on 40 observations, leading to 6.67 observations per predictor, lower than the 10 observations per predictor recommended by the Cochrane Handbook to prevent overfitting (itself a recommendation based on a rule of thumb for ordinary least squares regression) [29,30]. Moreover, the impact of seasonality (specifically, studies performed in winter) is numerically greater than that of training type: winter is associated with a serum vitamin D concentration lower by 8.586 ng/mL, almost double the 4.446 ng/mL impact of indoor/outdoor training.

Latitude is known to impact vitamin D levels due to different zenith angles and lengths of days, as well as local weather and cloudiness [82]. Some studies found an association between latitude and vitamin D levels [83,84], while others did not [85]. Still, diseases known to be correlated with vitamin D, such as multiple sclerosis [86], have also been shown to have a higher prevalence in northern countries [87].

Regarding gender, studies undertaken on the general population did not agree on whether there are significant differences between the vitamin D levels of females and males (one Chinese study found lower vitamin D levels in females compared to males [88], while another European study did not find such differences [89]; one study undertaken on COPD patients found vitamin D deficiency to be in connection with the male gender [90]). Although we believe some differences are to be expected due to biological as well as social causes [91], generally, information regarding gender differences was found to be lacking, especially in connection to occupational risks [8].

Specifically, subgroup analyses (Figure 3, Figure 4 and Figure 5) show that vitamin D levels are most different between indoor and outdoor training in the case of Asian athletes (in our case, athletes from Japan, Singapore, and South Korea). However, this observation needs to be confirmed by large-scale primary research. Another issue is the fact that self-declared race is an imperfect proxy for skin color, and populational genomics research suggests that alleles implicated in vitamin D metabolism, beyond skin color alone, vary according to ethnicity [92]. Finally, studies performed on African-American athletes were not included in a meta-regression model which controlled for race because of a very limited sample size compared to studies of Asian and Caucasian athletes.

The use of the season, rather than months, as a predictor has advantages and disadvantages. It has only four levels, which make it feasible for dummy variable regression, as it can be encoded by only three variables; however, it has low “resolution” in that it conflates observations that may be rather different (e.g., vitamin D nadir occurs in March, while May concentrations approach summer values [93,94,95], but all these observations would be labeled as “Spring”).

Of note is that specifics of the physical activities practiced by the athletes have to be taken into consideration. For example, clothing or other equipment worn during practice might decrease the exposed skin area, thus rendering it incapable of vitamin D synthesis. In the AusD study on the general population of four cities in Australia, the amount of skin exposed to the sun was the most important predictor of vitamin D levels, even more important than the latitude or season [84]. In yacht racing [96] or American football [97], for example, both the clothes and the equipment have to be adapted to allow for increased airflow, thus increasing the exposed skin areas. This would enable the athlete to cool down during effort. On the other hand, in colder climates, better insulation is required, thus the exposed skin area should be decreased [98]. This would further hinder the synthesis of vitamin D. Future analyses on this subject should take into consideration the type of sports practiced and their specifics.

Regardless of the exact interplay between ethnicity, geography, and seasonality on the one hand and the type of training on the other hand, a practical observation is that the difference between indoor and outdoor training is numerically small, and vitamin D sufficiency is far from universal in either group, especially in spring and winter. Specifically, if we consider the insufficiency threshold of 32 ng/mL used by Farrokhyar et al., we can observe that the random-effects pooled mean lies well below this value in spring and winter and is approximately equal to this threshold in autumn for both indoor and outdoor athletes, suggesting a clinically important prevalence of vitamin D insufficiency in either group during those seasons. In other words, we consider that testing for vitamin D levels and guiding supplementation accordingly should be considered regardless of the type of training, at least in the spring and winter months.

Methodologically, our paper has shown that meta-regression, especially multivariate meta-regression, is a viable technique in evaluating differences between subgroups in the presence of confounders, more so than simple subgroup analysis, and it shows potential in investigating mean differences between groups, starting from single-group studies, in the absence of comparative (primary) research.

Nonetheless, the limitations intrinsic to meta-analyses remain, namely their observational and post hoc character and, implicitly, their inability to infer causality. A limitation specific to meta-regression is its propensity to overfit, as well as its low power. While we managed to obtain a relatively large number of studies, its ability to correctly evaluate large models with multiple predictors is still limited. Our choice of confounding variables to analyze, namely race and season, has certain advantages and limitations as described above. Finally, while we tried to avoid the ecological fallacy, i.e., attributing group-level effects to individuals, its presence cannot be excluded entirely in any meta-regression.

## 5. Conclusions

We have identified that the type of training influences serum vitamin D concentrations, i.e., that outdoor training is associated with a slightly higher vitamin D concentration, at least when controlling for the season, latitude, and Asian/Caucasian race. This suggests that the impact of indoor/outdoor training is highly dependent on the presence of covariates; moreover, the impact of training type is of a rather low numerical and clinical magnitude. Likewise, vitamin D levels are significantly influenced by the season, i.e., the impact of winter is nearly double that of the training type. Consequently, a practical observation is that outdoor training alone is not sufficient in increasing vitamin D levels, and supplementation to correct insufficiency and deficiency should not be guided by the type of training alone.

On a theoretical level, we have proven that meta-regression can serve a role in filling gaps in the literature by investigating between-group differences when very few comparative studies exist, showing potential as a complement to primary observational studies.

## Figures and Tables

**Figure 1 ijms-24-07584-f001:**
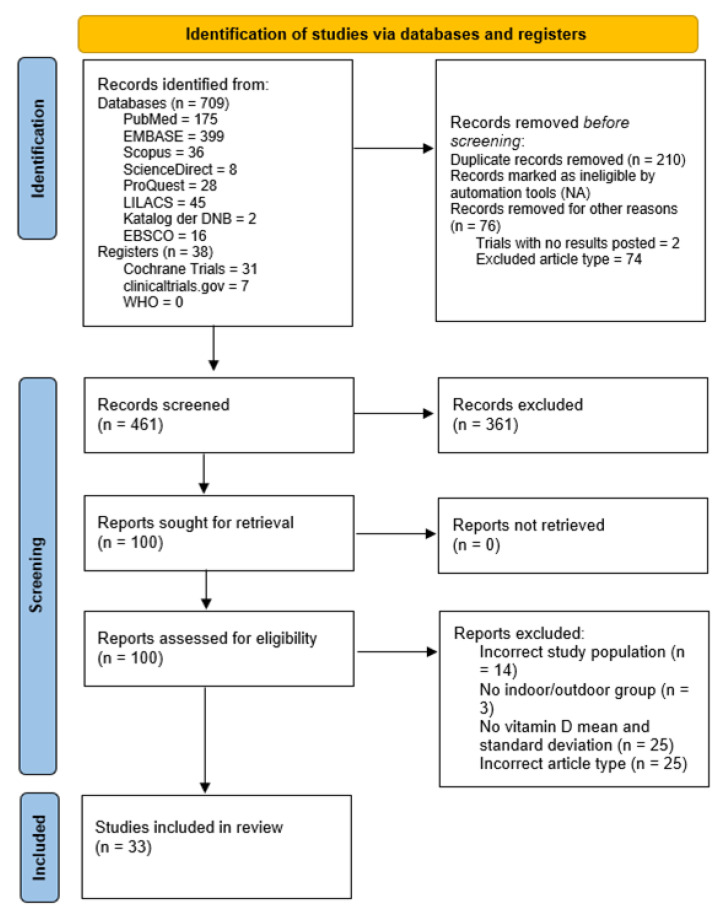
PRISMA flowchart for study selection.

**Figure 3 ijms-24-07584-f003:**
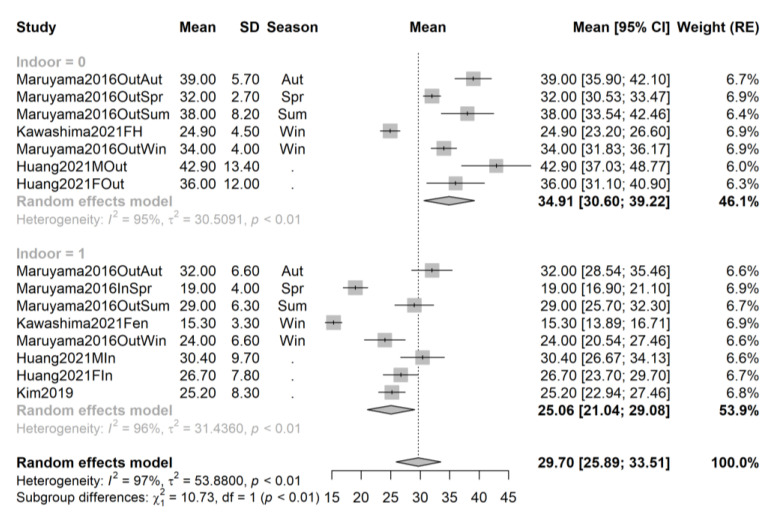
Forest plot of all studies performed on Asian athletes sorted by season. Studies [45,47,48,67].

**Table 1 ijms-24-07584-t001:** Characteristics of included studies. NR = not reported. Multiple = vitamin D sampled at multiple time points but not reported separately. Win = winter. Aut = autumn. Sum = summer. Spr = spring.

Study	Reference	Type	Risk of Bias	Sample Size	Male (%)	Age (Mean ± SD Where Available)	Season	Indoor/Outdoor	Country (Latitude)	Sport Type
Aydin, 2019	[40]	Cross-sectional	7 of 10	555	0.4126	15.9	Win–Aut	Both	Turkey (39° N)	Ballet, dance, defense sports, gymnastics, basketball, volleyball, athletics
Bauer, 2018	[41]	Cross-sectional	5 of 10	70	1	26.3 ± 4.9	Sum	Indoor	Germany (50° N)	Handball
Bauer, 2020	[42]	Cross-sectional	5 of 10	120	1	25.8 ± 5.2	Sum	Indoor	Germany (50° N)	Handball, ice hockey
Fields, 2020	[43]	Cross-sectional	5 of 10	36	0	19.4 ± 1.4	Win	Indoor	USA (38.8° N)	Track and field, basketball, volleyball
Geiker, 2017	[44]	Cross-sectional	4 of 10	29	0.586	18	Spr	Indoor	Denmark (56° N)	Swimming
Huang, 2021	[45]	Cross-sectional	5 of 10	95	0.4842	24.3 ± 4.3	Multiple	Both	Singapore (1° N)	15 sports
Jakse, 2019	[46]	Cross-sectional	4 of 10	31	0	16.6 ± 4.1	Spr	Indoor	Slovenia (46° N)	Swimming, gymnastics
Kawashima, 2021	[47]	Cross-sectional	5 of 10	48	1	19.8 ± 0.9	Win	Both	Japan (35° N)	Fencing, field hockey
Kim, 2019	[48]	Cross-sectional	4 of 10	52	1	23.8 ± 2.8	NR	Indoor	Taiwan (NR)	Volleyball
Ksiazek, 2018	[49]	Cross-sectional	3 of 10	25	1	21.9 ± 9.8	Win	Indoor	Poland (51° N)	Judo
Ksiazek, 2021	[50]	Cross-sectional	3 of 10	40	1	22.1 ± 3.4	Spr	Both	Poland (51° N)	Judo, football
McGill, 2014	[51]	Cross-sectional	5 of 10	108	NR	20.0 ± 1.15	Aut	Outdoor	USA (NR)	American football
Mehran, 2016	[52]	Cross-sectional	7 of 10	105	1	25.5 ± 4.4	Aut	Indoor	USA (NR)	Ice hockey
Most, 2021	[53]	Cross-sectional	5 of 10	112	1	26.1 ± 5.2	Sum	Indoor	Germany (50° N)	Handball, ice hockey
Peeling, 2012	[54]	Cross-sectional	6 of 10	72	0.597	16 ± 4	Sum	Both	Australia (32° S)	Gymnastics, diving, sailing, field hockey, athletics, rowing, water polo, sprint cycling
Radovanovic, 2022	[55]	Cross-sectional	5 of 10	18	0	21.2 ± 3.9	Aut	Indoor	Serbia (44° N)	Basketball
Ricart, 2021	[56]	Cross-sectional	5 of 10	27	0.481	15.8 ± 0.6	Win	Indoor	Spain (39.5° N)	Basketball
Sariakcali, 2020	[57]	Cross-sectional	3 of 10	36	1	23.3 ± 3.5	Sum	Outdoor	Turkey (40° N)	Football
Sghaier, 2015	[58]	Cross-sectional	5 of 10	150	0.616	18 ± 2	Win	Both	Tunisia (mean 35° N)	Athletics, judo, karate, boxing, fencing
Valtuena, 2014	[59]	Cross-sectional	3 of 10	408	0.583	22.8 ± 8.4	Multiple	Both	Spain (41.4° N)	34 sports
Wentz, 2016	[60]	Cross-sectional	6 of 10	59	0	23.5 ± 4.9	NR	Outdoor	USA (30.4° N)	Running
Caroli, 2014	[61]	Cohort	5 of 9	21	1	24.6 ± 4.3	Spr–Aut	Outdoor	Italy (44.9° N)	Rugby
Fields, 2019	[62]	Cohort	6 of 9	20	0.55	19.9 ± 1.2	Sum–Aut	Indoor	USA (38.8° N)	Basketball
Galan, 2012	[63]	Cohort	7 of 9	28	1	26.7 ± 3.6	Aut–Win	Outdoor	Spain (37.4° N)	Football
Haslacher, 2016	[64]	Cohort	8 of 9	47	0.915	65	NR	Outdoor	Austria (NR)	Marathoners, bicyclists
Huggins, 2019	[65]	Cohort	5 of 9	20	1	21 ± 1	Sum–Aut	Outdoor	USA (NR)	Soccer
Krzywanski, 2016	[66]	Cohort	4 of 9	409	0.557	25.1 ± 0.5	Spr–Sum–Win–Aut	Both	Poland (51.5° N)	Track and field, weightlifting, handball, volleyball
Maruyama, 2016	[67]	Cohort	7 of 9	27	0	20.6 ± 0.5	Spr–Sum–Win–Aut	Both	Japan (35° N)	Soccer, basketball, volleyball
Millward, 2020	[68]	Cohort	9 of 9	802	0.62	18.7 ± 1.2	Multiple	Both	USA (NR)	NCAA Division I student athletes
Wilson, 2020	[69]	Cohort	6 of 9	47	0.66	20.5 ± 1.7	Spr–Aut	Both	UK (51.2° N)	15 sports
Valtuena, 2021	[70]	Cohort	8 of 9	95	1	27.3 ± 4.6	Spr–Aut–Win	Both	Spain (41.4° N)	Football, indoor football, basketball, handball, roller hockey
Wyon, 2015	[71]	RCT	6 of 9	22	1	27.5 ± 9	Win	Indoor	UK (52.5° N)	Judo
Valenti, 2022	[72]	Before and after	7 of 12	8	1	47.5 ± 13.5	Sum	Outdoor	Italy (NR)	Cycling

**Table 2 ijms-24-07584-t002:** The results of bivariate meta-regressions for the following predictors of vitamin D levels: type of sport, gender, latitude, and Asian/Caucasian race. θ = intercept coefficient for each regression model. *β* = coefficient of the variable for each regression model. * = predictors with significant (*p* < 0.05) *β*-coefficient.

Variable (Coding)	Number of Included Studies	θ	*β* Estimate	*β* Standard Error	*β p*-Value (df1, df2)	R^2^ (%)
Type of sport * (indoor = 1)	91	31.387 (<0.001)	−3.692	1.843	0.045 (1, 91)	3.69
Gender (female = 1)	68	29.944 (<0.001)	2.511	2.126	0.237 (1, 66)	0.76
Latitude *	74	37.854 (<0.001)	−0.247	0.081	0.002 (1, 72)	10.52
Race (Caucasian = 1)	54	29.773 (<0.001)	−0.064	2.962	0.983 (1, 52)	<0.01

**Table 3 ijms-24-07584-t003:** Multivariate regression investigating the significance of season as a predictor of vitamin D levels. * = predictors with significant (*p* < 0.05) *β*-coefficient.

				F-Test for Moderators		
Coefficient	*β* Estimate	Standard Error	*p*-Value	F Statistic (df)	*p*-Value	R^2^ (%)
θ *	32.067	1.645	<0.001	28.605 (3)	<0.001	27.88
$\beta_{Spr}$ *	−8.761	2.461	<0.001			
$\beta_{Sum}$	2.051	2.575	0.426			
$\beta_{Win}$ *	−8.001	2.266	<0.001			

**Table 4 ijms-24-07584-t004:** Multivariate model investigating the significance of the type of sport, season, and latitude as predictors of vitamin D levels. * = predictors with significant (*p* < 0.05) *β*-coefficient.

				F-Test for Moderators			
Coefficient	*β* Estimate	Standard Error	*p*-Value	F Statistic (df)	*p*-Value	R^2^ (%)	Sample Size
θ *	46.470	5.852	<0.001	32.339 (5)	<0.001	31.06	66
Indoor	−1.262	1.807	0.485				
Spring *	−5.837	2.529	0.021				
Summer	3.398	2.696	0.207				
Winter *	−6.957	2.326	0.003				
Latitude *	−0.353	0.131	0.007				

**Table 5 ijms-24-07584-t005:** Multivariate model investigating the significance of the type of sport, season, latitude, and Asian/Caucasian race as predictors of vitamin D levels. * = predictors with significant (*p* < 0.05) *β*-coefficient.

				F-Test for Moderators			
Coefficient	*β* Estimate	Standard Error	*p*-Value	F Statistic (df)	*p*-Value	R^2^ (%)	Sample Size
θ *	39.995	7.791	<0.001	46.195 (5)	<0.001	52.98	40
Indoor *	−4.446	1.915	0.020				
Spring	−4.376	2.842	0.124				
Summer	4.804	2.695	0.075				
Winter *	−8.586	2.586	<0.001				
Latitude	−0.183	0.170	0.283				
Race (Asian = 1)	0.672	2.924	0.818				

## Data Availability

On request from the corresponding author.

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
