# Peer review of "Difference in Levels of Vitamin D between Indoor and Outdoor Athletes: A Systematic Review and Meta-Analysis"

_ijms, 2023, doi:10.3390/ijms24087584_

Round 1

Reviewer 1 Report

The presented review is addressing a very interesting topic. Looking into the Difference in levels of vitamin D between indoor and outdoor 2 athletes. Nice presentation of the results indicating that the type of sport is not an independent pre- dictor of vitamin D levels. The future is about preventing vitamin D insufficiency and deficiency, and that supplementation should not be guided on type of training alone. Data are clear and comprehensive.

Author Response

Dear Reviewer,

Thank you for your feedback!

In our subsequent changes, we have improved the overall readability of the paper.

Respectfully yours,

The Authors

Reviewer 2 Report

Dear Authors,

Please improve and extent your introduction

Please add more conclusion to your article

Kind regards

Author Response

Dear Reviewer,

Thank you for your feedback!
In our subsequent changes, we have improved the overall readability of the paper and have rewritten parts of the introduction, discussions and the conclusion to better represent our results

Respectfully yours,
The Authors

Reviewer 3 Report

The authors have presented an intriguing meta-analysis, and the utilization of meta-regression as an analytical approach is commendable. The overall methodology is sound, and the study adds value to the existing body of literature. However, I have a few suggestions that could potentially improve the paper's presentation and readability:

Tables 1.1 to 1.4 contain a significant amount of information, which can be overwhelming for readers. It might be beneficial to include these tables as supplementary material or to extract the most important details and incorporate them into the text.

Similarly, Table 2 would benefit from a more concise presentation. It would be helpful to summarize the information and emphasize the key findings within the text itself.

An interesting aspect that could be explored further is the relationship between vitamin D levels and the geographic regions in which the studies were conducted. A more in-depth discussion of this factor would provide valuable context for the study results.

Lastly, the paper could benefit from an expanded discussion on the potential impact of sport type and gender differences on the findings. This additional layer of analysis could provide valuable insights and help elucidate the factors that contribute to the observed effects.

Overall, the meta-analysis is well-executed, and the suggestions above could help improve the clarity and depth of the study.

Author Response

Dear Reviewer,

Thank you for your extensive feedback! We have moved the tables 1.1-1.4 and 2 to Supplementary data, and subsequently renumbered the remaining tables.

We have improved the readability of the text.

Moreover, we have added the following paragraphs to the Discussion chapter: paragraph 5 about latitude, paragraph 6 about gender and paragraph 9 about sport specifics.

Respectfully yours,

The Authors

Reviewer 4 Report

The present study sought to evaluate the difference between indoor and outdoor training in vitamin D concentration in athletes. The study is well conducted, and well written, with excellent statistical tools to evaluate the authors' hypothesis and transparency for all steps in the systematic review and meta-analysis.

This work can significantly contribute to the scientific knowledge of Sports Nutrition regarding potential factors affecting serum micronutrient concentration in athletes. I congratulate the authors for all the care conducting this research.

Author Response

(The authors gave the same response as above.)

Reviewer 5 Report

The authors analyzed vitamin D differences between indoor and outdoor athletes and reported that outdoor training is associated with higher vitamin D concentrations and overall vitamin D lack in athletes.

The authors did good work collecting the data and writing, but the methodology and reporting in the paper are inaccurate. This paper requires intensive work before being accepted:

1.       The table of baseline characteristics of the study and participants is missing. There should be a table showing the survey, the number of participants, avg age, country, Males, and females, etc., seasonal concentration, and or total concentration. So it is hard to judge the conclusion without knowing the sample size. Please check the paper that you cited and create similar tables like them (table 1 and table 2: DOI: 10.1007/s40279-014-0267-6)

2.       It would be better if the seasons were divided into two seasons only, autumn-winter or spring-summer. That would have made the paper more conscience. It is okay if they need to have four seasons, but the sample size should be reported in the tables as mentioned above.

3.       Results reported mainly regression, not the actual meta-analysis. You must report that outdoor has a marginal trend toward significance P=0.052) and write the numbers for both indoor and outdoor. Then you write the results of the regression.

4.       The conclusion of this study is wrong, especially in the abstract.

·         According to your research, the seasonal analysis did not impact vitamin D levels (in any season P>0.05).

·         You did not define the vitamin D insufficiency level range in your paper

·         You can say, “The impact of indoor or outdoor training is numerically small”, the rest of the sentence is inaccurate because there are other demographic parameters you did not study.

·         “While outdoor training is associated with higher vitamin D concentrations,” this was observed only in one racial group. Why did you generalize it?

·         “the prevalence of vitamin D insufficiency and deficiency is still significant among athletes” how did you know?

5.       Line 76: “Additionally, we performed subgroup analyses to account for the influence of gender, which season vitamin D was measured and latitude, among others” you did not perform subgroup for gender or latitude.

6.       Table 5.2. why mainly do you include Asian/Caucasian? Where is the African American?

7.       Aim stated that you measure indoor vs. outdoor. It would help if you mentioned your analysis for race and seasonal analysis in the introduction, not the methods.

 8.       Tables 1.1, 1.2. 1.3 can be summarized in one figure and moved data to supplementary.

9.       In all the paper, report only 2 decimal points, for example, 3.69 not 3.691, unless the P value has three digits, such as 0.001

10.   Table 2, move to supplementary.

11.   For the figures’ names, please use Figure 1, figure 2, instead of figure 1, figure 1.2, because the figures are not combined.

12.   Table 4. Multivariate regression should be a backward stepwise multivariate logistic regression analysis since Bsum is insignificant.

13.   Table 5.1, is the same. It should be a backward stepwise multivariate logistic regression analysis

14.   Line 354, PTH, write parathyroid hormone.

15.   Line 363: “to date, our meta-analysis is the first to quantify the differences in vitamin D levels between athletes training indoors or outdoors”. You have cited a previous meta-analysis that did the same, so please explain.

Author Response

Dear Reviewer,

We are grateful for your observations and have addressed them as follows:

1.The table of baseline characteristics of the study and participants is missing. There should be a table showing the survey, the number of participants, avg age, country, Males, and females, etc., seasonal concentration, and or total concentration. So it is hard to judge the conclusion without knowing the sample size. Please check the paper that you cited and create similar tables like them (table 1 and table 2: DOI: 10.1007/s40279-014-0267-6)

Thank you for your suggestion! We have created Table 1.

2.It would be better if the seasons were divided into two seasons only, autumn-winter or spring-summer. That would have made the paper more conscience. It is okay if they need to have four seasons, but the sample size should be reported in the tables as mentioned above.

We preferred to use 4 seasons because a two-season period would include months associated with drastically different vitamin D levels (e.g., vitamin D values are lowest in March and highest in August) and obscure the influence of seasonality.

3. Results reported mainly regression, not the actual meta-analysis. You must report that outdoor has a marginal trend toward significance P=0.052) and write the numbers for both indoor and outdoor. Then you write the results of the regression.

We expanded the section dedicated to the results of the meta-analysis (section 3.4. Pooling and forest plots), including the number of samples and studies, sample sizes, means and mean differences.

  1. The conclusion of this study is wrong, especially in the abstract.
  • According to your research, the seasonal analysis did not impact vitamin D levels (in any season P>0.05).

Indoor vs outdoor comparisons by means of subgroup analyses failed to achieve significance, both in the general case, and when considering each season separately. However, when including the seasons as independent variables in meta-regression models, at least one season achieves significance. We rewrote the Results section of the Abstract and the Conclusion to clarify this.

  • You did not define the vitamin D insufficiency level range in your paper

We added a paragraph referring to vitamin D insufficiency (with reference to Farrokhyar et al.) in the Discussion section (6th paragraph, page 19). References to insufficiency were removed from the Abstract because estimating the prevalence of insufficiency is not the objective of the paper.

  • You can say, “The impact of indoor or outdoor training is numerically small”, the rest of the sentence is inaccurate because there are other demographic parameters you did not study.

We rephrased the Results section of the Abstract completely, to better clarify which was the method used to estimate the impact of training type (subgroup analysis or meta-regression) and for which variables we controlled.

  • “While outdoor training is associated with higher vitamin D concentrations,” this was observed only in one racial group. Why did you generalize it?

Rephrased to remove ambiguity.

  • “the prevalence of vitamin D insufficiency and deficiency is still significant among athletes” how did you know?

Rephrased to avoid references to vitamin D insufficiency, the prevalence of which we did not estimate.

5. Line 76: “Additionally, we performed subgroup analyses to account for the influence of gender, which season vitamin D was measured and latitude, among others” you did not perform subgroup for gender or latitude.

Corrected (6th paragraph, page 2)

6. Table 5.2. why mainly do you include Asian/Caucasian? Where is the African American?

In order for an observation to be included in the model, it needs to have a value for all variables in the model. Of the two studies, McGill reports season but not gender, Millward reports gender but not season, and neither reports latitude. Since these are the only available studies on African-American athletes, we had no meaningful way to extrapolate these missing values; they were therefore removed from the analysis.

7. Aim stated that you measure indoor vs. outdoor. It would help if you mentioned your analysis for race and seasonal analysis in the introduction, not the methods.

We expanded the last paragraph of the Introduction to clarify this.

8. Tables 1.1, 1.2. 1.3 can be summarized in one figure and moved data to supplementary.

We have moved Tables 1.1-1.3 to Supplementary data.

9. In all the paper, report only 2 decimal points, for example, 3.69 not 3.691, unless the P value has three digits, such as 0.001

All in-text results other than p-values were truncated to 2 decimal points.

10. Table 2, move to supplementary.

Thank you for your suggestion, we have moved Table 2 to Supplementary data.

11. For the figures’ names, please use Figure 1, figure 2, instead of figure 1, figure 1.2, because the figures are not combined.

Thank you for your suggestion. We have renumbered the tables and figures.

12. Table 4. Multivariate regression should be a backward stepwise multivariate logistic regression analysis since Bsum is insignificant.

Thank you for your suggestion. We preferred to include all seasons as independent variables in the model to identify which ones are associated with a statistically significant change in vitamin D levels.

13. Table 5.1, is the same. It should be a backward stepwise multivariate logistic regression analysis

We built the model to take into account the predictors whose effects we were most interested in evaluating. Moreover, packages meta and metafor do not offer a convenient way to perform stepwise meta-regressions.

14. Line 354, PTH, write parathyroid hormone.

 Corrected

15. Line 363: “to date, our meta-analysis is the first to quantify the differences in vitamin D levels between athletes training indoors or outdoors”. You have cited a previous meta-analysis that did the same, so please explain.

The paragraph was rewritten to clarify how our study complements to that of Farrokhyar et al.

Respectfully yours,

The Authors